# Non-Invasive Diagnostic Techniques in Penile Intraepithelial Neoplasia (PeIN): Insights from Reflectance Confocal Microscopy (RCM), Line-Field Confocal Optical Coherence Tomography (LC-OCT), and Correlation with Histopathological Features

**DOI:** 10.3390/dermatopathology12030019

**Published:** 2025-07-07

**Authors:** Caterina Damiani, Cesare Ariasi, Giuseppe La Rosa, Francesca Di Lauro, Mariachiara Arisi, Vincenzo Maione, Marina Venturini, Simone Soglia

**Affiliations:** Department of Dermatology, University of Brescia, 25121 Brescia, Italy

**Keywords:** PeIN, LC-OCT, RCM, 3D, skin cancer, non-invasive diagnostics

## Abstract

Penile intraepithelial neoplasia (PeIN) is a rare but clinically significant condition that can progress to invasive squamous carcinoma. Early diagnosis is crucial but often challenging due to its heterogeneous clinical and dermoscopic presentation, which can mimic other benign or malignant lesions. In this study, we report two cases of pigmented penile lesions evaluated using non-invasive imaging techniques: reflectance confocal microscopy (RCM) and line-field confocal optical coherence tomography (LC-OCT). Both methods revealed characteristic features such as hyperkeratosis, parakeratosis, acanthosis, nuclear pleomorphism of keratinocytes, and the presence of bright intraepithelial dendritic cells, correlating closely with histopathological findings of high-grade basaloid PeIN. Our findings highlight the valuable role of RCM and LC-OCT in improving the differential diagnosis of genital lesions, potentially reducing the need for invasive diagnostic procedures and ensuring early, appropriate management.

## 1. Introduction

Non-invasive imaging techniques have become increasingly important in dermatological diagnostics, offering real-time, high-resolution evaluation of skin lesions without the need for biopsy. Among these, reflectance confocal microscopy (RCM) and line-field confocal optical coherence tomography (LC-OCT) are two advanced modalities that allow in vivo assessment of cutaneous structures at nearly histological resolution [1].

RCM provides en face images of the epidermis and superficial dermis with excellent cellular detail, making it particularly useful for assessing epidermal and superficial dermal structures. However, its limited penetration depth (typically up to 200–250 μm) restricts its utility in evaluating deeper components of the skin, and its purely horizontal imaging orientation may complicate the interpretation of certain architectural features [2]. It works by directing a low-power near-infrared laser (typically at 830 nm) onto a single point within the skin. The light reflected from that focal point is captured through a pinhole that excludes out-of-focus light, resulting in high-contrast, horizontal optical sections. By scanning across the tissue point by point and reconstructing the signal, RCM generates detailed grayscale images of cellular and architectural structures, primarily based on differences in refractive indices. Structures with higher refractive indices, such as melanin, keratin, and inflammatory cells, appear brighter, allowing precise morphological evaluation of the epidermis, dermoepidermal junction, and superficial dermis [3]. LC-OCT combines the optical principles of conventional optical coherence tomography (OCT) and RCM. OCT is a technique that uses low-coherence interferometry to obtain high-resolution, cross-sectional images of biological tissues. A broadband light source is split into two beams: one directed toward the tissue and the other toward a reference mirror. Light reflected from both paths is recombined to produce interference patterns only when the optical path lengths match within the coherence length. By analyzing these interference signals, OCT reconstructs depth-resolved images of the tissue (B-scans). The technique provides micrometer-scale resolution and millimeter-scale penetration, making it especially useful for imaging layered structures like the skin, retina, and mucosa. Image contrast arises from differences in the tissue’s optical scattering properties [4]. The combination of the two methods allows the acquisition of both vertical (cross-sectional) and horizontal (en face) images of the skin with high spatial resolution. With a penetration depth of up to approximately 500 μm, LC-OCT allows for a detailed, three-dimensional representation of the epidermis and superficial dermis [5]. It combines both vertical and horizontal imaging, enabling quasi-histological visualization of tissue architecture in real time [1]. These techniques have shown promising results in the diagnosis and monitoring of various skin cancers and inflammatory dermatoses [5,6,7,8,9,10,11]. However, their application in rare or anatomically challenging conditions, such as penile intraepithelial neoplasia (PeIN), remains underexplored.

PeIN is a rare but clinically significant condition that encompasses a spectrum of premalignant lesions affecting the penile epithelium, previously classified pathologically as squamous cell carcinoma in situ. PeIN has the potential to progress to invasive penile carcinoma, with reported progression rates ranging from approximately 2.6% to 14%. These data underscore the importance of early diagnosis and timely management to reduce the risk of malignant transformation [12,13].

Clinically, PeIN includes three main variants: erythroplasia of Queyrat (EQ), Bowen disease (BD), and bowenoid papulosis (BP), which differ in terms of anatomical distribution, epidemiology, clinical presentation, and potential for malignant transformation.

EQ typically affects the glans and prepuce of older, uncircumcised men, presenting as a red, velvety plaque with a relatively high risk of progression (10–33%) and frequent association with high-risk human papillomavirus (HPV) types, particularly 8 and 16 [14,15].

BD arises on the penile shaft, appearing as a scaly, crusted red plaque, with a lower risk of invasive transformation (~5%) and often linked to HPV-16 [16,17].

BP occurs mainly in younger men as multiple reddish-violet papules. Although histologically indistinguishable from PeIN, it generally follows a benign course, though progression can occur, especially in older or immunosuppressed individuals [16,18]. It is also strongly associated with HPV-16 [19].

Despite their clinical distinctions, these entities are histopathologically indistinguishable and uniformly diagnosed as PeIN.

Histologically, it is characterized by intraepithelial neoplastic proliferation with variable degrees of dysplasia, keratinization, and nuclear atypia, without extension beyond the basement membrane. It is classified into non-HPV-related (differentiated) and HPV-related (warty, basaloid, and warty–basaloid) subtypes. Basaloid PeIN is the predominant type and preferentially associated with HPV in about 76–99% of cases. HPV-16 is the most frequently detected genotype in basaloid PeIN [20].

While HPV-associated PeIN is driven by the oncogenic effects of high-risk HPV infection, HPV-independent PeIN is primarily associated with chronic inflammatory dermatoses, particularly lichen sclerosus (LS) [21].

Differentiated PeIN affects typically elderly men with inflammatory lesions, most often LS, and a common site is the foreskin. Undifferentiated PeIN affects younger men, and the preferred site is the glans. PeIN lesions are often multifocal and heterogeneous, especially warty/basaloid subtypes [22].

Differentiated PeIN clinically presents as white or pink macules or plaques, while undifferentiated PeIN presents as flat or slightly elevated lesions that are velvety, erythematous, or dark brown, with irregular or sharply delineated borders. The only clinical and dermoscopic evaluation is insufficient for a certain diagnosis, as PeIN (especially the basaloid subtype) can be mistaken for entities such as urothelial carcinoma in situ, lichen planus-like keratosis, seborreic keratosis, and melanoma [23]. Other possible differential diagnoses include LS, lichen planus, psoriasis, genital warts, Zoon balanitis, and fixed drug eruption.

Among the currently available non-invasive diagnostic techniques, dermoscopy is the most widely used in clinical practice for the evaluation of PeIN. Although its cellular resolution is lower compared to advanced imaging modalities such as RCM and LC-OCT, dermoscopy remains a practical and accessible tool. Characteristic dermoscopic features of PeIN include structureless pink areas and a polymorphous vascular pattern, with dotted, glomerular, and linear vessels frequently arranged in clusters. In pigmented variants—such as Bowenoid papulosis and pigmented Bowen disease—brown to grey dots and globules may be seen, often in linear or mosaic arrangements [24]. Despite the absence of a pigment network, these features can raise suspicion of PeIN and guide the selection of biopsy sites. While dermoscopy lacks the cellular-level resolution provided by RCM and LC-OCT, it continues to play a supportive role in the non-invasive assessment and surveillance of penile lesions.

## 2. Materials and Methods

Two male patients, aged 75 and 65 years, presented to the Dermatology Department of the University of Brescia with asymptomatic pigmented lesions located on the shaft of the penis. The first patient reported the presence of the lesion for approximately 18 months, while in the second case, the lesions had been present for more than 2 years. Both lesions were flat, with well-defined margins, and were clinically evaluated using dermoscopy (10× magnification) (Figure 1a,b). In both cases, the lesions were asymptomatic, and the patients had no personal or family history of cutaneous tumors.

To further investigate the lesions, non-invasive imaging techniques were employed. Reflectance confocal microscopy (RCM) was performed using the VivaScope 3000 device (Caliber Imaging and Diagnostics, Rochester, NY, USA), while line-field confocal optical coherence tomography (LC-OCT) imaging was conducted with the DeepLive™ LC-OCT system (DAMAE Medical, Paris, France). Both examinations provided high-resolution, cellular-level imaging of the lesions (Figure 2a,d,e).

Following non-invasive imaging, punch biopsies were performed on both lesions to obtain histopathological confirmation. Hematoxylin–eosin staining and immunohistochemical analysis with p16 and Ki67 markers were conducted. HPV testing was also performed. Histological findings were then correlated with imaging results to validate the non-invasive diagnostic observations (Figure 2b,e).

The study was conducted in accordance with the Declaration of Helsinki. All patients were given verbal and written information on the nature of the study, and they signed an informed consent form before enrollment.

## 3. Results

Dermoscopy of both penile lesions revealed a predominant ring-like pattern, a blue-whitish veil, shiny white structures, and isolated thin scales (Figure 1a,b). Non-invasive imaging with RCM and LC-OCT revealed consistent features across both cases. At the epidermal level, hyperkeratosis and parakeratosis were evident, along with marked acanthosis and elongation of the rete ridges (Figure 2a,d,e). RCM imaging highlighted the presence of keratinocyte nuclear pleomorphism and numerous bright intraepithelial dendritic cells (Figure 2d). LC-OCT provided both vertical and three-dimensional projections, revealing at the epidermal level the presence of large bright atypical cells with dendritic projections and confirming the undulating yet well-preserved dermo-epidermal junction, along with a superficial dermis enriched with melanophages and inflammatory infiltrate (Figure 2a,e).

Histopathological examination confirmed the imaging findings, showing a monotonous population of small anaplastic basaloid cells with marked basal atypia, numerous mitoses, and elongated rete ridges. Immunohistochemical analysis revealed full-thickness positivity for p16 and suprabasal positivity for Ki67, while HPV testing was negative in both cases (Figure 2b,e). These findings were compatible with a diagnosis of high-grade basaloid PeIN. Following diagnosis, both lesions were treated with complete surgical excision.

## 4. Discussion

PeIN represents a challenging diagnostic entity due to its variable clinical presentation and the potential overlap with both benign and malignant lesions. In particular, pigmented lesions of the genital area can easily be misdiagnosed, leading either to unnecessary invasive procedures or to delayed treatment of malignant conditions. Our study highlights the value of integrating non-invasive imaging techniques, such as RCM and LC-OCT, in the diagnostic workflow for suspected PeIN.

The use of non-invasive imaging techniques such as LC-OCT and RCM allows for detailed visualization of the cellular morphology and stands of high interest even in genital and mucosal sites, where surgery can be invasive or even destruent [25].

Both RCM and LC-OCT provided a highly detailed, non-invasive assessment of key architectural and cytological features associated with PeIN. These features included hyperkeratosis, parakeratosis, acanthosis, nuclear pleomorphism of keratinocytes, and elongation of the rete ridges—all of which are hallmarks of intraepithelial neoplastic processes. One notable observation was the presence of bright, dendritic-appearing cells within the epidermis. While such a finding may initially raise suspicion for a melanocytic lesion, the diagnostic pathway was refined through the absence of atypical melanocytic nests and the presence of an atypical honeycomb pattern—features that are more consistent with keratinocytic rather than melanocytic pathology. This comprehensive multimodal imaging approach enabled a more accurate and confident differentiation between PeIN and a variety of clinically and dermoscopically similar conditions, such as seborrheic keratosis, invasive squamous cell carcinoma, melanocytic nevi, and melanoma. [1,6,7]. Importantly, the structural and cellular-level imaging features identified by RCM and LC-OCT closely mirrored those seen on histopathological examination, thereby reinforcing the diagnostic reliability and clinical utility of these advanced imaging modalities in the non-invasive evaluation and management of PeIN.

The accurate differentiation of PeIN from other entities is critical, particularly in the sensitive anatomical context of the genital area, where surgical procedures may be complex and have significant morbidity. Non-invasive imaging allowed us to better define the lesion margins, reduce diagnostic uncertainty, and plan appropriate surgical management, ultimately improving patient outcomes.

In literature, several reports have described the application of RCM and LC-OCT to the evaluation of genital lesions. These non-invasive imaging techniques have been used to study a variety of conditions affecting the genital mucosa, including genital warts, molluscum contagiosum, and pigmented lesions [25,26,27]. Both RCM and LC-OCT have demonstrated the ability to visualize microstructural features with near-histological resolution, aiding in the diagnosis and differential diagnosis of these entities. However, limitations remain—such as the difficulty in detecting specific cytological features [26].

Our experience aligns with recent literature emphasizing the role of LC-OCT and RCM in dermatological oncology, especially in mucosal and genital sites, where traditional biopsy may be associated with higher risk of complications. These techniques offer a real-time, painless alternative for initial lesion assessment and monitoring over time, potentially reducing the need for repeated biopsies.

However, despite their advantages, non-invasive imaging methods are not yet a substitute for histopathological confirmation, particularly when dealing with high-grade intraepithelial neoplasias. Future larger-scale studies are warranted to further validate specific imaging criteria for PeIN and to establish standardized diagnostic algorithms incorporating these advanced technologies. A major limitation of our study is the small sample size, as it includes only two patients.

The management of PeIN aims to completely eradicate the lesion while minimizing functional and cosmetic sequelae. Therapeutic options include both topical and surgical modalities, with the choice depending on lesion size, location, multifocality, and patient-related factors. Topical agents such as imiquimod 5% cream and 5-fluorouracil are typically considered for small, well-demarcated lesions and offer the advantage of tissue preservation [28]. However, they require prolonged treatment courses and close clinical follow-up due to variable response rates and the potential for recurrence. In this context, non-invasive imaging techniques such as RCM and LC-OCT are emerging as valuable tools for monitoring therapeutic response, allowing real-time visualization of architectural and cellular changes without the need for repeated biopsies.

Surgical management remains the gold standard for extensive, multifocal, or recurrent lesions, offering high cure rates and histopathologic confirmation of complete excision. In our case series, both patients with histologically confirmed PeIN underwent surgical excision of the lesions, with clear margins. The procedures were well tolerated and resulted in excellent cosmetic outcomes, with no evidence of recurrence during follow-up. Among surgical options, wide local excision, glans resurfacing, and, in select cases, Mohs micrographic surgery are commonly employed, balancing oncologic control with organ preservation [13,28]. Circumcision is often indicated when lesions involve or are confined to the foreskin. Regardless of the chosen treatment, long-term surveillance is crucial given the risk of local recurrence and progression to invasive squamous cell carcinoma.

In addition to conventional surgical and topical therapies, alternative and adjunctive treatment modalities have been explored for PeIN, particularly in cases where tissue preservation is prioritized or recurrences occur. Photodynamic therapy (PDT), which involves the application of a photosensitizing agent followed by exposure to a specific wavelength of light, has shown some promise in the management of PeIN by inducing selective cytotoxicity in dysplastic cells. PDT offers the advantages of being non-invasive and tissue-sparing, with favorable cosmetic outcomes [29]. However, its efficacy appears to be variable, and recurrence rates remain a concern, particularly in high-grade or multifocal lesions. Moreover, treatment-related discomfort and the need for multiple sessions may limit patient compliance. Emerging combined approaches, such as the use of topical therapies (e.g., imiquimod or 5-fluorouracil) in conjunction with prophylactic or therapeutic HPV vaccination, are under investigation [30]. These strategies aim not only to treat the lesion but also to modulate the underlying viral-driven pathogenesis, especially in HPV-related PeIN. Preliminary reports suggest a potential synergistic effect in terms of lesion regression and recurrence prevention. However, robust data specifically on the effectiveness of HPV vaccination as a treatment for existing PeIN are limited, and most guidelines still emphasize its preventive use [31]. Further research is needed to establish efficacy, optimal timing, and patient selection criteria. If validated, these approaches may represent valuable additions to the therapeutic armamentarium for PeIN, particularly in patients seeking conservative or organ-sparing alternatives.

## 5. Conclusions

The advancement of imaging technologies, particularly the most recent LC-OCT, is expected to significantly enhance diagnostic efficiency in the future. LC-OCT enables real-time visualization of skin structures with near-histological resolution, representing a form of “digital biopsy” that may be increasingly employed both for diagnostic purposes and for therapeutic monitoring and follow-up of PeIN, as well as other cutaneous lesions.

In conclusion, our findings suggest that RCM and LC-OCT can play a pivotal role in the early and non-invasive diagnosis of PeIN, aiding clinicians in distinguishing it from other pigmented lesions and supporting more tailored and effective patient management strategies.

## Figures and Tables

**Figure 1 dermatopathology-12-00019-f001:**
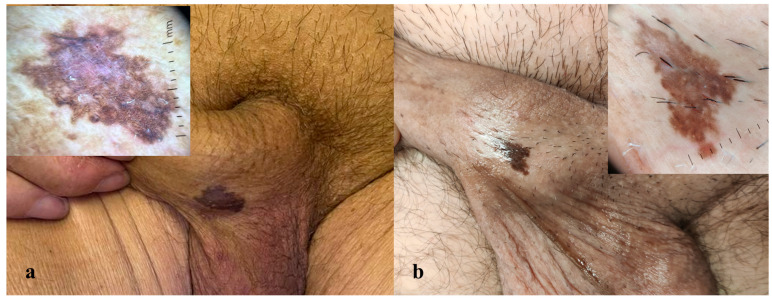
(**a**,**b**) Macroscopic images of two clinically similar pigmented lesions on the penile shaft, each with the corresponding dermoscopic image (10×). Dermoscopy reveals a predominant ring-like pattern, a blue-whitish veil, and shiny white structures in both cases. Isolated thin scales are present in dermoscopy (**a**).

**Figure 2 dermatopathology-12-00019-f002:**
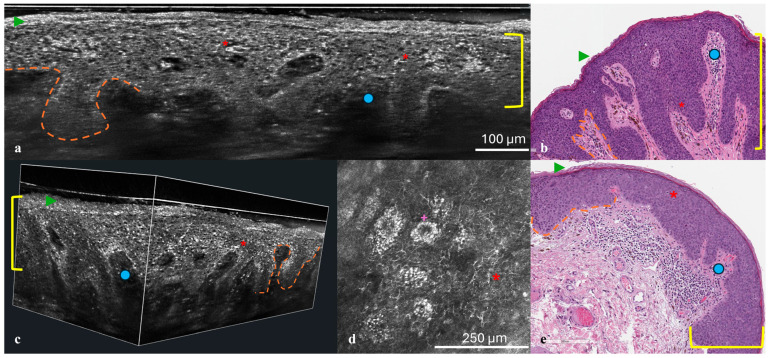
(**a**–**e**) Histological correlation with RCM and LC-OCT. Vertical (**a**) and 3D (**c**) LC-OCT projection, histologic presentation (**b**,**e**), and RCM at the level of the stratum spinosum (**d**). At the epidermal level, compact hyperkeratosis (green arrow) and acanthosis (yellow bracket) are observed, along with keratinocyte nuclear pleomorphism with scattered intraepithelial bright dendritic melanocytes (red asterisk). The dermo-epidermal junction appears well preserved (orange dashed line), with elongation of the papillae (blue dot). In the RCM image, bright keratinocytes are evident (purple cross).

## Data Availability

The data that support the findings of this study are available from the corresponding author upon reasonable request.

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
