# Peer review of "Non-Invasive Diagnostic Techniques in Penile Intraepithelial Neoplasia (PeIN): Insights from Reflectance Confocal Microscopy (RCM), Line-Field Confocal Optical Coherence Tomography (LC-OCT), and Correlation with Histopathological Features"

_dermatopathology, 2025, doi:10.3390/dermatopathology12030019_

Round 1

Reviewer 1 Report

Comments and Suggestions for Authors

Dont you believe Dermoscopy alone is useful  to give good diagnostic orientation and to help the dermatologist to a guided punch biopsy? The new techniques of imaging are not alwais present in the hospital.No matter of debate about the risk, very low indeed, of biopsy, but the gold standard is histology after biopsy addressed by techniques of imaging. The old dermoscopy is still useful, as in these cases, to suspect a malignant lesion and to addresse a surgical removal.

Author Response

Comment 1: Dont you believe Dermoscopy alone is useful  to give good diagnostic orientation and to help the dermatologist to a guided punch biopsy? The new techniques of imaging are not alwais present in the hospital.No matter of debate about the risk, very low indeed, of biopsy, but the gold standard is histology after biopsy addressed by techniques of imaging. The old dermoscopy is still useful, as in these cases, to suspect a malignant lesion and to addresse a surgical removal

Answer 1: Dermoscopy remains a valuable first-line tool in the diagnostic process of cutaneous lesions. While newer imaging techniques (such as reflectance confocal microscopy or optical coherence tomography) offer additional detail, they are not universally available, especially in many hospital settings. In contrast, dermoscopy is widely accessible, non-invasive, and cost-effective, and it plays a crucial role in guiding clinical decisions. Importantly, dermoscopy can provide strong diagnostic orientation, helping the dermatologist to select the most appropriate area for a targeted punch biopsy. This is particularly relevant when the lesion is clinically ambiguous or heterogeneous.

Non-invasive diagnostic techniques are rapidly evolving, and in the coming years, their costs are expected to decrease while the performance of the devices continues to improve. Currently, they help guide diagnosis and are poised to play an increasingly important role in non-invasive monitoring.

In our case, the lesions were surgically removed; however, if a topical treatment—such as 5% Imiquimod—were chosen instead, advanced non-invasive diagnostics would allow for a better assessment of therapeutic success and improved monitoring of potential recurrence over time. More invasive methods, such as biopsy, could then be reserved as a second-line option if needed.

Reviewer 2 Report

Comments and Suggestions for Authors

I have reviewed the report: "Non-invasive diagnostic in penile intraepithelial neoplasia (PeIN): insights from reflectance confocal microscopy (RCM), line field confocal optical coherence tomography (LC-OCT) and correlation with histopathological features." I have the following suggestions:

  1. The introduction part talks about PeIN. It should also give a brief introduction to reflectance confocal microscopy (RCM) and line field confocal optical coherence tomography (LC-OCT).
  2. In line 78 it is stated that p16 was positive while HPV testing was negative. Can the authors describe the type of HPV testing that was carried out?
  3. In the discussion part, can the authors discuss their results in comparison with existing data/publications on RCM and LC-OCT in PeIN.
  4. Only two cases are presented; can the authors add that this may be a limitation? Thank you.

Author Response

We would like to thank the reviewer for the thoughtful and constructive comments. Please find below our responses to each point raised:

1. "The introduction part talks about PeIN. It should also give a brief introduction to reflectance confocal microscopy (RCM) and line-field confocal optical coherence tomography (LC-OCT)."

We appreciate this suggestion. A brief introduction to both RCM and LC-OCT has been added to the revised Introduction section.

2. "In line 78 it is stated that p16 was positive while HPV testing was negative. Can the authors describe the type of HPV testing that was carried out?"

Thank you for this observation. Unfortunately, we were unable to retrieve specific information regarding the HPV testing kit used, as the analysis was performed by our pathology service and the details were not available in the patient records. 

3. "In the discussion part, can the authors discuss their results in comparison with existing data/publications on RCM and LC-OCT in PeIN?"

Thank you for pointing this out. To our knowledge, there are currently no published studies specifically addressing the use of RCM or LC-OCT in PeIN. We have clarified this in the Discussion section and emphasized the novelty of our observations in this context.

4. "Only two cases are presented; can the authors add that this may be a limitation?"

We agree with the reviewer. A sentence acknowledging the limited number of cases as a study limitation has been added to the Discussion section.

We are grateful for the reviewer’s comments, which have helped us improve the quality and clarity of our manuscript.

Reviewer 3 Report

Comments and Suggestions for Authors

I read with great interest the manuscript. However, I have some comments:

-It is important to include the clinical classification of PEIN : 3 clinically distinct variants constitute most cases of penile intraepithelial neoplasia – Bowen disease, erythroplasia of Queyrat and bowenoid papulosis...--> differences  in epidemiology/aetiological associations/prognosis… then as you mention histopathologically, these variants may be indistinguishable and show SCC in situ which can be further divided into differentiated and undifferentiated subtypes… also, it may be helpful to note that HPV-associated PeIN is linked to the oncogenic effects of HPV, whereas HPV-independent PeIN typically arises in the context of chronic inflammation. These two pathways show distinct histopathological characteristics, which could be emphasized to aid in differential diagnosis.

-line 43: in the differential diagnosis of PEIN should be included and other genital dermatoses such as lichen sclerosus,
lichen planus, psoriasis, genital warts, Zoon balanitis and fixed
drug eruption

-just a clarification…the clinical and dermoscopic photos in fig 1(a,b) are from the 2 cases you describe(aged 75 and 65 years)?

-In fig 1b, dermoscopically there is no blue-whitish veil nor thin scales.

-lines 78-79: highlight what % of the cases of basaloid PEIN HPV testing are usually positive and negative. Also mention which HPV variants are usually found. you could put it in the discussion section

- I believe that including a concise table summarizing the key RCM and LC-OCT features of PEIN and its main differential diagnoses would greatly aid clinicians in making accurate comparisons and reaching a diagnosis more efficiently.

-Also, is there a way to differentiate with RCM and/or LC-OCT the subtypes of PEIN? If yes, please highlight some hints for each subtype.

Overall, this article is a solid effort. With some revisions, it can be greatly enhanced.

Author Response

We sincerely thank the reviewer for the insightful and constructive comments, which have helped improve the clarity and clinical relevance of our manuscript. Please find below our detailed responses to each point:

1. “It is important to include the clinical classification of PEIN: 3 clinically distinct variants constitute most cases of penile intraepithelial neoplasia – Bowen disease, erythroplasia of Queyrat and bowenoid papulosis... differences in epidemiology/aetiological associations/prognosis…”

We agree with the reviewer and have now added a paragraph in the Introduction section summarizing the three main clinical variants of PeIN (Bowen disease, erythroplasia of Queyrat, and bowenoid papulosis), including their differences in epidemiology, aetiology, and prognosis. We also clarified that, histopathologically, these variants may appear indistinguishable and correspond to squamous cell carcinoma in situ, which can be further subclassified into differentiated and undifferentiated types.

2. “It may be helpful to note that HPV-associated PeIN is linked to the oncogenic effects of HPV, whereas HPV-independent PeIN typically arises in the context of chronic inflammation…”

Thank you for this suggestion. We have incorporated a paragraph into the Introduction and expanded in the Discussion to emphasize the distinction between HPV-associated and HPV-independent PeIN.

3. “Line 43: in the differential diagnosis of PEIN should be included other genital dermatoses such as lichen sclerosus, lichen planus, psoriasis, genital warts, Zoon balanitis and fixed drug eruption.”

We have revised the text accordingly and included the suggested conditions in the differential diagnosis of PeIN.

4. “Just a clarification… the clinical and dermoscopic photos in Fig. 1(a,b) are from the 2 cases you describe (aged 75 and 65 years)?”

Yes, we confirm that the clinical and dermoscopic images shown in Figure 1 (a and b) correspond to the two cases described (patients aged 75 and 65 years, respectively). 

5. “In Fig. 1b, dermoscopically there is no blue-whitish veil nor thin scales.”

Thank you for this observation. We have revised the legend to Figure 1b to more accurately reflect the dermoscopic findings. The blue-whitish veil is indeed present in the upper portion of the lesion; however, we acknowledge that it may not be clearly appreciable due to the limitations in image resolution and size. Regarding the thin scales, we confirm that desquamation is not observed in this case, and we have corrected this error accordingly in the revised figure legend.

6. “Lines 78–79: highlight what % of the cases of basaloid PeIN HPV testing are usually positive and negative. Also mention which HPV variants are usually found. You could put it in the introduction section.”

We appreciate this valuable suggestion. In the Discussion section, we have added data from the literature regarding HPV positivity rates in basaloid PeIN (typically high, >80%), and the most frequently associated high-risk HPV types, particularly HPV 16.

7. “I believe that including a concise table summarizing the key RCM and LC-OCT features of PEIN and its main differential diagnoses would greatly aid clinicians in making accurate comparisons and reaching a diagnosis more efficiently.”

We appreciate the reviewer’s thoughtful suggestion. While we agree that such a table could be helpful in a clinical context, we believe that including it in the present manuscript—based on only two cases—might overstate the generalizability of our observations and risk suggesting a recurring diagnostic pattern that has not yet been validated in a larger cohort. In our view, this could inadvertently lead to premature conclusions. Therefore, we would prefer not to include the table at this stage. However, should the editorial board consider it essential, we remain available to prepare and include a concise version.

8. “Also, is there a way to differentiate with RCM and/or LC-OCT the subtypes of PEIN? If yes, please highlight some hints for each subtype.”

Thank you for this insightful question. At present, it is not possible to reliably differentiate the histopathological subtypes of PeIN using RCM or LC-OCT. In our study, we included only two cases, and to our knowledge, there are no existing data in the literature that provide a systematic comparison of imaging patterns across the different PeIN subtypes. A larger case series would be necessary to identify potential subtype-specific features and to determine whether distinct RCM or LC-OCT patterns can be consistently associated with HPV-associated versus HPV-independent PeIN.

We once again thank the reviewer for the helpful feedback that has strengthened our manuscript. All suggested revisions have been incorporated, and the text has been carefully edited for clarity and precision.

Round 2

Reviewer 3 Report

Comments and Suggestions for Authors

The authors have taken my suggestions into account and the manuscript has improved as a result.